# Assessment of Telehealth Literacy in Users: Survey and Analysis of Demographic and Behavioral Determinants

**DOI:** 10.3390/healthcare13151825

**Published:** 2025-07-26

**Authors:** Marcela Hechenleitner-Carvallo, Jacqueline Ibarra-Peso, Sergio V. Flores

**Affiliations:** 1Departamento de Ciencias Básicas y Morfológicas, Facultad de Medicina, Universidad Católica de la Santísima Concepción, Concepción 4090541, Chile; marcelahc@ucsc.cl; 2Oficina de Educación en Ciencias de la Salud, Facultad de Medicina, Universidad Católica de la Santísima Concepción, Concepción 4090541, Chile; 3Observatorio Regional de Salud Digital, CRT Biobío, Concepción 4030000, Chile; jibarra@ucsc.cl; 4Departamento de Ciencias Clínicas y Preclínicas, Facultad de Medicina, Universidad Católica de la Santísima Concepción, Concepción 4090541, Chile; 5Vicerrectoría de Investigación e Innovación, Universidad Arturo Prat, Iquique 1110939, Chile

**Keywords:** telehealth literacy, health disparities, digital health, predictive models, survey validation

## Abstract

**Background**: Telehealth is an essential component of modern healthcare, and it was especially relevant during the COVID-19 pandemic, but disparities in digital and technological literacy among health professionals may limit its equitable adoption and impact. **Objective**: This study seeks to validate an eight-item telehealth literacy survey among health professionals in Central–South Chile and to examine demographic and behavioral determinants of literacy levels, developing predictive models to identify key factors. **Methods**: In this cross-sectional study, 2182 health professionals from urban and rural centers in Central–South Chile completed the adapted survey along with questions on age, gender, nationality, and frequency of telehealth use. We assessed internal consistency (Cronbach’s α), explored factor structure via exploratory factor analysis (EFA), and tested associations using Pearson correlations, *t*-tests, one-way ANOVA, and both linear and multinomial logistic regressions. **Results**: The instrument demonstrated high reliability (Cronbach’s α = 0.92) and a two-factor structure explaining 65% of variance. Age negatively correlated with literacy (r = −0.26; *p* < 0.001), while the frequency of telehealth use showed a positive correlation (r = 0.26; *p* < 0.001). Female professionals and those in urban settings scored significantly higher on telehealth literacy (*p* = 0.005 and *p* < 0.001, respectively). The reduced multinomial model achieved moderate classification accuracy (51.65%) in distinguishing low, medium, and high literacy groups. **Conclusions**: The validated survey is a reliable tool for assessing telehealth literacy among health professionals in Chile. The findings highlight age, gender, and geographic disparities, and support targeted digital literacy interventions to promote equitable telehealth practice.

## 1. Introduction

The accelerated adoption of telehealth during the COVID-19 pandemic has profoundly transformed healthcare delivery by offering innovative solutions to overcome geographic, economic, and accessibility barriers. However, this digital transformation has also revealed profound inequalities associated with demographic, technological, and digital health literacy factors. In this context, telehealth literacy has emerged as a critical competency for equitable access to healthcare, especially within increasingly digitized healthcare systems.

Telehealth literacy encompasses a multidimensional set of skills that are needed to interact effectively with digital health platforms. It includes not only technical skills, but also cognitive and communicative competencies embedded in specific social contexts [1]. This approach has supported a broader conceptualization that recognizes the interactions between individual skills, structural conditions, and social determinants of health.

Recent studies have shown that factors such as age, language proficiency, race, gender, and socioeconomic status significantly influence access, quality, and satisfaction with telehealth services [2,3,4]. For example, African American patients, Medicare users, and people with limited English proficiency show lower telehealth adoption rates and tend to prefer telephone consultations to video calls, which may restrict the quality of care they receive [2]. Similarly, patients with lower levels of health literacy or income tend to perceive virtual consultations as less useful, and report difficulty recalling clinical information provided during remote visits [3].

These challenges are particularly evident among older adults, who, despite being heavy users of healthcare services, often face substantial limitations in digital skills, making it difficult for them to participate effectively in telehealth settings [5,6]. Key barriers for this group include low digital confidence, limited training in the use of technology, and infrastructure issues such as unreliable Internet connectivity [7]. In these cases, social support and caregiver involvement are crucial to the successful adoption of telehealth tools [7,8].

To address these issues, multiple interventions have been proposed to improve digital health literacy among older populations. While these interventions show promising results, there is still limited evidence regarding their impact on concrete clinical outcomes [6]. In parallel, assessment tools such as the eHEALS, the most commonly used questionnaire for both adults and children [9], and TLST [8] have been developed to identify individuals at risk of digital exclusion, allowing for more targeted interventions.

Telehealth literacy gaps are also determined by cultural and contextual factors. Research conducted in Saudi Arabia and in underserved Hispanic communities in the United States shows that sociodemographic characteristics, access to technology, and trust in healthcare providers affect people’s willingness to use digital platforms [7,10]. Similarly, in regions such as sub-Saharan Africa, structural barriers, such as lack of support for implementation and concerns about privacy and confidentiality, underscore the need for culturally tailored approaches [11].

Addressing these inequities requires a systemic response. A multi-stakeholder strategy has been proposed, including policy reforms, clinician training, and models of care co-designed with user communities [12]. Previous experiences have shown that strengthening the capacities of providers in rural areas through distance education programs can effectively bridge the digital divide, especially when content is tailored to the specific needs of each user group [13].

During public health emergencies such as the COVID-19 pandemic, the rapid deployment of telehealth solutions highlighted both the opportunities and systemic shortcomings of national health systems. In Peru, for example, the experience highlighted the need for regulatory flexibility, infrastructure investment, and proactive digital inclusion policies to effectively integrate telehealth into the national health system [14]. At the institutional level, Schwamm [15] proposed a strategic framework for redesigning care pathways to sustainably incorporate telehealth into routine clinical services.

Another critical aspect is the design of culturally and linguistically appropriate services. Tailoring services to the diversity of patient populations—along with the involvement of trusted intermediaries, such as community health workers and local organizations—has proven effective in improving the uptake of telehealth among vulnerable populations [16,17]. In addition, technological support, including access to digital devices and reliable connectivity, is essential for socioeconomically disadvantaged groups [18]. The most successful initiatives combine these components with targeted digital literacy programs, resulting in more effective and sustained service adoption [19,20].

Similarly, methodological tools such as telehealth equity panels allow for the systematic tracking of disparities in access and use, facilitating continuous improvement of services [21]. These approaches have proven particularly effective in programs targeting children with special health care needs (Van [22,23], people living with HIV [20,24], and survivors of sexual assault in rural areas [17].

Finally, there is an urgent need for professional training in telehealth. Qualitative studies suggest that many clinical teams faced significant challenges during the pandemic due to inadequate preparation for remote care [25,26]. In parallel, ensuring adequate clinical documentation in telehealth settings remains a crucial challenge to maintaining quality of care [27].

Taken together, the literature demonstrates that telehealth literacy is a new social determinant of equitable access to health care. Its complexity demands comprehensive and culturally sensitive strategies. However, significant gaps remain, both in the availability of validated assessment tools and in understanding the sociodemographic and behavioral determinants of telehealth literacy. In response to these challenges, the present study aims to validate a brief survey to assess telehealth literacy and examine its associations with demographic and use-related variables. The ultimate goal is to generate evidence to inform future interventions, public policies, and implementation strategies that reduce inequities in digital health settings.

## 2. Materials and Methods

In this cross-sectional study, data were collected from a sample of 2182 participants recruited using a convenience sampling strategy from a population from South–Central Chile. Convenience sampling is a non-probability method that selects participants based on ease of access; it is fast, inexpensive, and simple to implement for exploratory research, but it prevents the calculation of sampling errors, introduces selection and self-selection biases, and limits external validity and generalizability. Recruitment took place during health promotion campaigns in both urban and rural health centers, allowing for geographic diversity despite the non-probability nature of the sample. We recruited participants through on-site visits to primary health care centers. Research staff approached eligible individuals attending these centers and invited them to take part in the study. Eligibility criteria included individuals over 18 years of age who had previous contact with telehealth services and were able to give informed consent. Efforts were made to include participants from diverse backgrounds, nationalities, and socioeconomic levels. The survey was administered and collected in person by trained research assistants. All assistants underwent standardized training in survey administration procedures and ethical participation protocols.

An eight-item telehealth literacy survey developed by Norman et al. [28] and later applied in Spain [29,30] was administered to assess the access to, the understanding of, and the ability to evaluate telehealth services. Each item was rated on a 5-point Likert-type scale, with higher scores indicating higher literacy. Validation of the instrument, in the Spanish version [29,30] including reliability and construct validity analyses, was conducted as part of this study. In addition to survey responses, demographic variables (age, sex, nationality, and urban or rural residence) and behavioral data (frequency of telehealth use) were collected.

All statistical analyses were performed using the R software (version 4.3.1), including the packages “psych”, “dplyr”, “tidyverse” and “factoextra”, “psc1”, “mice”, “missForest”, and “nnet”. The analytical workflow was divided into three phases. In Phase 1, validation of the survey instrument was carried out. For this purpose, reliability was assessed using Cronbach’s alpha coefficient, considering a value above 0.70 acceptable, and item–total correlations were calculated in order to determine the individual contribution of each item. Likewise, construct validity was explored through an exploratory factor analysis (EFA) to identify the latent dimensions underlying the instrument, using the Tucker–Lewis index (TLI) and the root mean square error of approximation (RMSEA) as fit indicators. Exploratory factor analysis (EFA) is a multivariate technique that uncovers underlying latent factors explaining the correlation structure among observed variables, thus reducing dimensionality and validating theoretical constructs.

Regarding the treatment of missing data, the items presented between 7 and 18 missing values, which were imputed by means of the mean to maintain a complete data set in the subsequent analyses. To confirm that mean imputation did not distort our findings, we compared internal consistency (Cronbach’s α) and factorial structure (the first two eigenvalues) across three imputation methods: item-wise mean, missForest (random forests), and multiple imputation by chained equations (MICE) with five imputations using predictive mean matching. Cronbach’s α ranged from 0.9177 to 0.9179, and the first two eigenvalues differed by less than 0.004 units, confirming that the simplest approach (mean imputation) fully preserves the reliability and latent structure of the questionnaire.

During Phase 2, correlations were examined and comparisons between groups were made. Pearson correlation coefficients were calculated to assess associations between continuous variables, such as age, telehealth literacy scores, and frequency of use. To analyze differences in literacy scores by demographic characteristics (e.g., sex and the type of residence), independent *t*-tests and one-way ANOVAs were employed. Where normality assumptions were violated, the nonparametric Kruskal–Wallis test was used. In addition, Chi-square tests were used to explore associations between categorical variables, such as nationality and literacy level, applying caution in the interpretation of low-frequency categories.

In Phase 3, we began by estimating a full multiple linear regression to predict each participant’s continuous telehealth literacy score from age (years), sex (0 = female, 1 = male, 2 = other), frequency of telehealth use (total number of visits), rural residence (1 = rural, 0 = urban), and six nationality indicator variables (Bolivian, Chilean [reference], Colombian, Mexican, Peruvian, or Venezuelan). We examined unstandardized coefficients, standard errors, t-values, *p*-values, and overall fit (R^2^, adjusted R^2^, F-statistic), then compared this full model to a reduced four-predictor specification (omitting all nationality dummies) via changes in AIC, BIC, adjusted R^2^, and a block F-test on the change in residual sum of squares. Because including nationality yielded only trivial fit gains and risked unstable estimates with our small immigrant subsample, we adopted the more parsimonious model with age, sex, frequency, and rural residence for our primary continuous analyses.

Next, we converted total literacy scores into three groups—low (≤25th percentile), medium (25th–75th), and high (≥75th)—to ensure roughly balanced cell sizes and fitted a multinomial logistic regression predicting these categories from the same initial six predictors. Although the outcome variable is ordinal, we fitted a multinomial logistic model to avoid potential violations of the proportional-odds assumption. This choice may modestly reduce statistical power, yet it does not change the direction or substantive interpretation of the findings. As before, we compared the full vs. reduced (no-nationality) specifications using AIC, BIC, and a likelihood–ratio test on the block of nationality variables; finding little improvement from nationality, we retained the four-predictor multinomial model (reference outcome = low) for our categorical analyses. Finally, we assessed classification performance via a confusion matrix and overall accuracy to quantify how well age, sex, telehealth frequency, and rural residence distinguished literacy levels.

This study received approval from the Ethics Committee of the Universidad Católica de la Santísima Concepción N° 60/2022, in accordance with the ethical guidelines for research involving human subjects. Written informed consent was obtained from all participants. All data were anonymized prior to analysis to ensure confidentiality and data protection.

## 3. Results

In total, 2202 participants were surveyed. The mean age was 44.1 years (SD = 16.85), with a range between 18 and 98 years. The mean telehealth literacy score was 23.76 (SD = 10.03), out of a possible 8–40 points (sum of eight items rated on a 5-point Likert scale), while the frequency of telehealth use showed a low mean (1.70 ± 4.01). In terms of sociodemographic variables, 61.2% of participants identified themselves as female, 38.8% as male, and 0.05% as “other”. In terms of residence, the majority lived in urban areas (79.3%), while 20.7% came from rural areas. The sample was mostly composed of Chilean nationals (96%), with minority participation of citizens from Venezuela, Peru, Argentina, Bolivia, Colombia, and Mexico, each with less than 1% representation (Table 1). No language barriers were detected, as all participants were Spanish speakers.

### 3.1. Reliability Analysis

Internal consistency was strong (Cronbach’s α = 0.92), indicating that the eight items form a highly coherent scale. Item–total correlations ranged from 0.58 to 0.79, demonstrating that every item makes a meaningful contribution to the overall score. Moreover, deleting any single item led to negligible changes in α (range after deletion: 0.90–0.92), which confirms the stability and robustness of the instrument.

### 3.2. Construct Validity

Prior to conducting exploratory factor analysis (EFA), sampling adequacy was confirmed by a Kaiser–Meyer–Olkin measure of 0.92 and a significant Bartlett’s test of sphericity (χ^2^(28) = 11 330.8, *p* < 0.001), indicating that the correlation matrix was well suited for factorization. Parallel analysis suggested up to three factors, but, in the interest of parsimony and theoretical interpretability, a two-factor solution was retained. This solution explained 65% of the total variance and exhibited clear factor loadings: items 1–3 loaded strongly on Factor 1 (conceptual familiarity), while items 4–8 loaded on Factor 2 (practical application). Both the root-mean-square error of approximation (RMSEA = 0.077) and the Tucker–Lewis index (TLI = 0.967) indicated excellent fit relative to a null model. In general, TLI values above 0.95 are considered to reflect good fit, suggesting that the proposed factor structure explains the observed data substantially better than a null model.

The analysis identified two factors that together explained 65% of the total variance (35% for the first factor and 30% for the second factor). The first factor mainly grouped items p2 and p3, with loadings above 0.70, and reflected functional and technical skills related to the use of telehealth. The second factor included high loadings on items p5, p6, and p7 (≥0.69), associated with the critical evaluation of digital information and confidence in its application. The full factor loadings, together with communalities (h^2^), uniqueness (u^2^), and levels of complexity, are presented in Table 2.

The fit indices confirmed the adequacy of the model, with a Tucker–Lewis index (TLI) of 0.969, a root mean square error of approximation (RMSEA) of 0.076, and a significant goodness-of-fit test (χ^2^(13) = 176.9; *p* < 0.001). Taken together, these findings support the two-dimensional structure of the instrument and suggest good construct validity, although confirmatory factor analyses are recommended in future studies to strengthen the proposed model.

### 3.3. Analysis Between Variables

Pearson’s correlation analysis showed statistically significant associations between continuous variables, though all effect sizes were small. Age was negatively related to telehealth literacy scores (r = −0.27, *p* < 0.001), indicating a weak inverse relationship, and the frequency of telehealth use was positively correlated with literacy (r = 0.26, *p* < 0.001), also reflecting a small effect. The association between age and frequency of telehealth use was negligible (r = −0.049, *p* = 0.024), suggesting virtually no practical relationship despite statistical significance.

When examining differences between demographic groups, it was found that women had significantly higher telehealth literacy scores (M = 24.11) than men (M = 22.86), a difference that was statistically significant (t(2180) = −2.81, *p* = 0.005, 95% CI [−2.13, −0.38]). Likewise, participants residing in urban areas had higher literacy (M = 24.15) compared to those living in rural areas (M = 21.72), a difference that was also significant (t(2180) = 4.76, *p* < 0.0001; 95% CI [1.43, 3.43]).

Regarding associations between categorical variables, no significant relationship was observed between sex and the type of residence, according to the Chi-square test (χ^2^ = 0.003, *p* = 0.956). In addition, no significant association was found between nationality and the level of telehealth literacy (χ^2^ = 22.23, *p* = 0.39). This association was corroborated by a Kruskal–Wallis test, which also evidenced significant differences between mean scores by nationality (χ^2^ = 22.28, df = 13, *p* = 0.0022). The mean literacy scores by nationality are presented in Table 3.

### 3.4. Association Between Independent Variables and Literacy Levels

Significant differences were observed in age and frequency of telehealth use according to literacy levels. Analysis of variance (ANOVA) showed that both factors varied significantly between the different literacy groups. In the case of age, a significant difference was found with F(3, 2091) = 176.2 and a *p* value less than 2 × 10^−16^. Similarly, the frequency of telehealth use also presented significant differences, with F(3, 2094) = 194.4 and *p* < 2 × 10^−16^.

These findings were confirmed by nonparametric Kruskal–Wallis tests, which also indicated statistically significant differences in both age (χ^2^ = 1857.1, gl = 3, *p* < 2.2 × 10^−16^) and frequency of telehealth use (χ^2^ = 310.83, gl = 3, *p* < 2.2 × 10^−16^). In line with these results, Figure 1 presents a boxplot showing that participants with prior telehealth experience tend to have higher literacy scores compared to those who have not had such experience. Taken together, these results suggest that older age and less exposure to telehealth services are associated with lower levels of telehealth literacy.

### 3.5. Regression Analysis

We first fitted a multiple linear regression (Table 4) that predicted the continuous telehealth literacy score from age (years), sex (1 = female, 0 = male), “Frequency using telehealth” (number of telehealth visits), rural residence (1 = rural, 0 = urban), and nationality indicators (BOL = Bolivian; CHI = Chilean [reference]; COL = Colombian; MEX = Mexican; PER = Peruvian; VEN = Venezuelan). Age remained a significant negative predictor (β = −0.143, t(2085) = −11.71, *p* < 0.001), and “Frequency using telehealth” retained a strong positive association (β = 0.607, t(2085) = 11.88, *p* < 0.001). The Mexican nationality dummy also reached significance, with Mexican respondents scoring on average 28.8 points lower than Chileans (β = −28.78, t(2085) = −2.67, *p* = 0.008); none of the other nationality dummies (Bolivian, Colombian, Peruvian, or Venezuelan) reached significance. Sex (β = 0.762, *p* = 0.069) and rural residence (β = −0.918, *p* = 0.071) showed marginal trends, and the intercept was 37.41 (*p* < 0.001). These ten predictors together explain 13.9% of the variance (R^2^ = 0.139, adjusted R^2^ = 0.135, F(10, 2085) = 33.68, *p* < 0.001).

When comparing the full model—including nationality dummies—with a reduced model without nationality dummies, we found that adding nationality slightly improved AIC (15 321.0 vs. 15 324.4) and yielded a nominally significant block test (ΔRSS = 1 338.5, F(6,2085) = 2.565, *p* = 0.0177). However, BIC favored the simpler model (15 388.8 vs. 15 358.3), and the gain in the adjusted R^2^ was negligible (0.135 vs. 0.131). Given the very small number of immigrant participants—and the resulting risk of biased, unstable estimates—we therefore prioritized a more parsimonious specification without nationality for our primary analyses.

As shown in Table 5, the final linear regression that predicted continuous telehealth literacy scores from age, sex (1 = male), frequency of telehealth use, and residence (1 = rural) yielded the following significant effects: the intercept was 28.83 (SE = 0.65, t = 44.22, *p* < 2 × 10^−16^); age was negatively associated (β = −0.1452, SE = 0.012, t = −11.87, *p* < 2 × 10^−16^); “Frequency of telehealth” use was strongly positive (β = 0.603, SE = 0.051, t = 11.80, *p* < 2 × 10^−16^); sex showed a modest male advantage (β = 0.8470, SE = 0.418, t = 2.02, *p* = 0.043); and rural residence tended toward lower literacy rates (β = −0.981, SE = 0.509, t = −1.93, *p* = 0.054).

The final reduced regression equation is as follows:Ŷᵢ = 28.82646 − 0.14526·Ageᵢ + 0.84702·Sexᵢ + 0.60381·Frequency of telehealthᵢ − 0.98165·Ruralᵢ
where Ŷᵢ = predicted the telehealth literacy score for individual I; Ageᵢ = age in years; Sexᵢ = 1 if male, 0 if female; Frequency of telehealthᵢ = number of telehealth visits; and Ruralᵢ = 1 if rural, 0 if urban.

We next fitted a multinomial logistic regression (Table 6) to classify participants into low, medium, and high telehealth literacy levels using the same predictors: age (years), sex (1 = male, 0 = female), frequency using telehealth, rural residence (1 = rural, 0 = urban), and nationality indicators (BOL = Bolivian; CHI = Chilean [reference]; COL = Colombian; MEX = Mexican; PER = Peruvian; VEN = Venezuelan).

The frequency of telehealth use emerged as the strongest predictor—each additional visit increased the odds of high versus low scores by 92% (OR = 1.92, 95% CI [1.31, 2.82], *p* < 0.001) and medium versus low scores by 68% (OR = 1.68, [1.14, 2.46], *p* = 0.008). Rural residence markedly reduced odds in both contrasts (high: OR = 0.06, [0.03, 0.13], *p* < 0.001; medium: OR = 0.08, [0.04, 0.18], *p* < 0.001). Age had a small negative effect only for high versus low scores (OR = 0.976 per year, *p* = 0.010), sex was non-significant (both *p* > 0.80), and nationality effects were unstable with large standard errors.

We compared the full multinomial model—including nationality dummies—with a reduced model omitting them and found that the reduced specification had substantially lower AIC (4286.9 vs. 4296.1) and BIC (4371.6 vs. 4482.4), and the likelihood–ratio test for the block of nationality variables was non-significant (Δ Deviance = 26.81 on 18 df, *p* = 0.083). Given these results—and the small number of immigrant participants, which can lead to unstable nationality estimates—we therefore adopted the more parsimonious multinomial model without nationality for our primary analyses. This decision converged with our linear regression strategy, where nationality indicators were likewise dropped to ensure model simplicity and stability.

We refitted the multinomial logistic regression without nationality indicators to produce a more parsimonious model (Table 7). The reduced log-odds equations (low as reference) are as follows:log(P(Highᵢ|Xᵢ)/P(Lowᵢ|Xᵢ)) = 4.8367 − 0.02495·Ageᵢ + 0.02557·Sexᵢ + 0.65019·Frequency of telehealthᵢ − 2.81311·Ruralᵢlog(P(Mediumᵢ|Xᵢ)/P(Lowᵢ|Xᵢ)) = 3.6399 + 0.00328·Ageᵢ − 0.07724·Sexᵢ + 0.51299·Frequency of telehealthᵢ − 2.49069·Ruralᵢ

As shown in Table 7, age was a small negative predictor of high versus low literacy (OR = 0.976, *p* = 0.010) and non-significant for medium literacy; sex was non-significant in both contrasts (*p* > 0.80); the frequency of telehealth use was a significant positive predictor (high: OR = 1.916, *p* = 0.001; medium: OR = 1.670, *p* = 0.009); and rural residence was a significant negative predictor in both comparisons (high: OR = 0.060, *p* < 0.001; medium: OR = 0.083, *p* < 0.001).

### 3.6. Predictive Model Evaluation

We evaluated the reduced multinomial model (without nationality) using a confusion matrix (Table 8). The overall accuracy was 51.7%. The sensitivity was 16.7% for the low-literacy group, 37.8% for the medium-literacy group, and 75.9% for the high-literacy group; specificity was 92.6%, 74.9%, and 50.7%, respectively. Misclassification was especially pronounced in the medium category: only 271 of 717 medium-literacy cases were correctly identified (37.8%), with 51.9% were misassigned to high-literacy and 10.3% to low-literacy. This pattern shows that the model reliably distinguishes the extremes but struggles with intermediate literacy levels. Future work should consider threshold adjustments or additional predictors to bolster discrimination in the medium range.

## 4. Discussion

The results provide valuable information on factors influencing telehealth literacy, with significant implications for improving access to and participation in telehealth services.

### 4.1. The Reliability and Validity of the Survey Instrument

The telehealth literacy survey demonstrated excellent internal consistency, with a Cronbach’s alpha of 0.92, confirming its reliability in measuring the intended construct. Exploratory factor analyses identified two underlying dimensions that explained 65% of the variance: specific literacy skills and confidence or critical skills in navigating health information. These results are in line with theoretical frameworks in health literacy research. Haun et al. [31] conducted a comprehensive review of 51 health literacy measurement tools and found that most assessments do not capture all dimensions of health literacy, particularly the digital literacy components essential for telehealth adoption. Their findings support the need for comprehensive assessment tools that measure both general health literacy and digital competencies.

In addition, the World Health Organization [32] has stressed the importance of strengthening digital health literacy as a fundamental component of global health strategies, ensuring that populations can access and effectively use telehealth services to improve health outcomes.

### 4.2. Associations Between Independent Variables and Literacy Levels

Age and literacy: A negative association was observed between age and literacy levels, with older participants showing lower literacy levels. Shi [33] highlighted that older adults face significant barriers to adopting digital health tools due to limited prior exposure, usability issues, and lack of tailored training programs. Their study demonstrated that digital health interventions, without age-specific support, risk widening health disparities rather than reducing them. Furthermore, Stellefson et al. [34] suggest that digital health interventions should integrate usability improvements to accommodate older populations, reducing the risk of digital exclusion.

Telehealth use and literacy: A positive correlation between frequent telehealth use and higher levels of literacy suggests a bidirectional relationship, where greater exposure improves literacy and vice versa. Van der Vaart and Drossaert [35] developed and validated the Digital Health Literacy Instrument (DHLI), identifying key digital literacy skills needed for effective telehealth use, such as navigation skills, information evaluation, and privacy protection. Their findings emphasize the importance of integrating digital literacy training into healthcare services to improve patient engagement. In addition, Greenhalgh et al. [36] argue that digital health interventions should be designed with user experience in mind, ensuring accessibility for diverse populations.

Gender differences: Women scored significantly higher in telehealth literacy compared to men. Stellefson et al. [34] conducted a systematic review of digital health literacy among college students and found that women tend to use online health information more frequently, demonstrating better information-seeking strategies and greater trust in digital health platforms. Their findings suggest that gender differences in digital health behaviors should be taken into account when designing interventions. Similarly, Seckin et al. [37] indicate that women may be more proactive in chronic disease management through telehealth tools, reinforcing the need for gender-sensitive approaches to digital health initiatives. One possible explanation for higher telehealth literacy among women is that they more frequently engage with health services—particularly in maternal and newborn care—where remote consultations have become common. Also, women often take primary responsibility for family health and may therefore develop greater familiarity with telehealth platforms through prenatal visits, pediatric check-ins, and follow-up care. Future studies should explore gendered patterns of telehealth use across different clinical contexts to confirm these hypotheses.

Urban–rural disparities: A significant gap in literacy levels was found between urban and rural participants, emphasizing the need for policy interventions to reduce inequalities in access to and the utilization of telehealth services. Greenhalgh et al. [36] analyzed telehealth implementation in different settings and found that infrastructure constraints, provider resistance, and sociocultural factors contributed to lower adoption rates in rural areas. Their study underscores the need for tailored telehealth policies that address logistical and technological barriers in underserved communities. The World Health Organization [32] has also highlighted the importance of digital health literacy programs in rural and low-resource settings to improve healthcare accessibility and reduce disparities. Urban residents’ higher telehealth literacy scores may reflect greater access to reliable internet and digital devices—as well as more frequent exposure to telehealth services through well-resourced clinics and hospitals. In urban areas, higher average levels of education and digital familiarity can facilitate comfort with online platforms, user interfaces, and troubleshooting that contribute to stronger eHealth skills. Conversely, rural residents often face connectivity challenges, fewer local telehealth offerings, and less opportunity for hands-on guidance, all of which can limit both uptake and confidence in using remote health technologies.

Patient-centered digital health literacy: Research highlights the importance of patient-centered care in digital health interventions. Seckin et al. [37] explored the impact of eHealth literacy on chronic disease management and found that higher literacy levels were associated with better self-management, medication adherence, and patient–provider communication. Their study suggests that improving eHealth literacy can improve chronic disease outcomes by empowering patients to take a more active role in their healthcare decisions. In addition, the World Health Organization [32] has stressed that digital health strategies should be patient-centered, integrating information from diverse populations to optimize telehealth interventions.

In our predominantly urban Chilean primary-care sample, age showed a modest negative correlation with telehealth literacy (r = −0.27) but only a negligible link to actual usage (r = −0.05). This gap suggests that although older adults felt less comfortable and familiar with telehealth platforms, they still used them—likely driven by greater chronic-care needs and direct encouragement from local clinics. In contrast, younger participants had higher digital fluency but fewer health-related reasons to log on, so their strong literacy did not translate into more visits. These findings imply that, even when access initiatives boost adoption, targeted usability support remains essential to help older patients leverage telehealth effectively.

### 4.3. Predictive Model Performance

The model achieved moderate overall accuracy (51.7%) and performed well at the extremes—high-literacy cases were identified with 75.9% sensitivity and low-literacy cases with 92.6% specificity—but it struggled with intermediate proficiency: only 37.8% of medium-literacy participants were correctly classified (with over half misassigned to the high-literacy group). This imbalance suggests the current predictors capture extreme telehealth literacy levels but lack nuance for the middle range. Future work should explore threshold adjustments or include additional covariates (e.g., prior telehealth experience, educational attainment) to bolster discrimination among medium-literacy users.

### 4.4. Implications for Practice and Policy

The results suggest the need for specific educational interventions tailored to specific demographic groups and the strengthening of telehealth infrastructure in rural areas. In addition, integrating digital literacy into routine medical visits could improve patient engagement with these services.

Given the identified disparities in telehealth access by age, gender, and geographic location, a tailored approach to telehealth programs is recommended. The application of predictive models could help identify individuals with low literacy levels and prioritize them for targeted interventions, consistent with the principles of equity and patient-centered care.

Our study makes three clear contributions to telehealth research. First, we locally validated and applied a telehealth literacy survey adapted for Chilean users, addressing the gap left by prior work focused mostly on North America and Europe. Second, we demonstrated that medical necessity can sustain telehealth use among older adults even when their confidence with digital tools is lower, showing that “need” and “skill” do not always move in parallel. Third, by examining both literacy and usage across gender, urban versus rural residence, and nationality, we offer concrete guidance on where to direct training efforts and infrastructure investments. Together, these insights confirm earlier findings on age-related digital health challenges while adding practical evidence on how real-world needs shape telehealth adoption.

### 4.5. Limitations

This study has some limitations, such as reliance on self-reported data and its application in a specific geographic context, which may affect generalizability. Future research should validate the survey in diverse populations, explore the impact of telehealth literacy on clinical outcomes and patient satisfaction, and consider adding relevant variables—such as digital access barriers, socioeconomic status, or health condition severity—to the instrument to enhance its explanatory power. The World Health Organization [32] has emphasized the need to develop global strategies to support digital health literacy initiatives, especially in low-resource settings.

## 5. Conclusions

Using a validated telehealth literacy instrument, we identified two dimensions—conceptual familiarity and practical application—with strong reliability. Regression analyses revealed younger age, higher telehealth use frequency, and urban residence as significant literacy predictors. The model achieved moderate accuracy across levels. These findings highlight the need for targeted interventions such as digital literacy training, primary care digital readiness assessments, and community workshops to strengthen telehealth skills among older, rural, and infrequent telehealth users in Chile, which would improve access and reduce health disparities.

## Figures and Tables

**Figure 1 healthcare-13-01825-f001:**
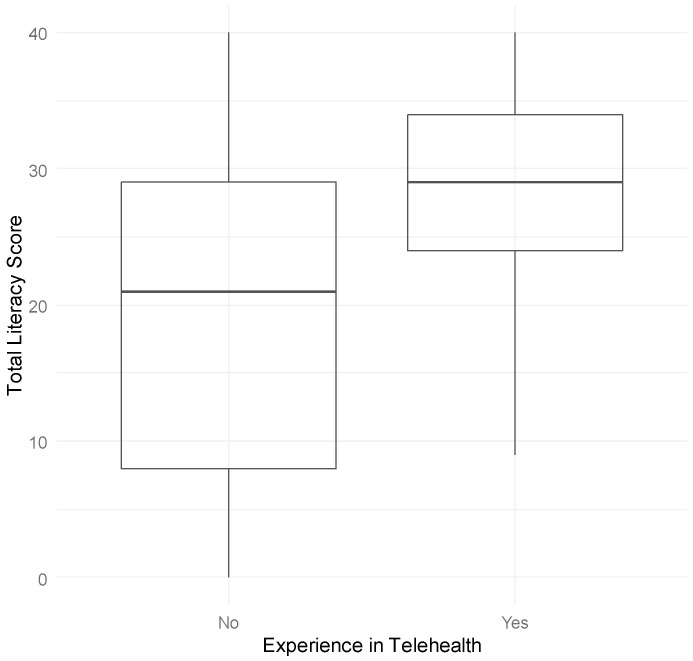
Distribution of telehealth literacy scores according to previous telehealth experience. Telehealth literacy scores range from 8 (minimum) to 40 (maximum) points (sum of eight items rated 1–5).

**Table 1 healthcare-13-01825-t001:** Descriptive characteristics of the study sample.

Category	Variable/Level	Value
Numerical variables	Age (mean ± SD)	44.11 ± 16.85
	Ages	12–98
	Telehealth use (mean ± SD)	1.70 ± 4.01
	Literacy score (mean ± SD)	23.76 ± 10.03
Sex (%)	Woman	61.2%
	Male	38.8%
	Other/No answer	0.05%
Residence (%)	Urban	79.34%
	Rural	20.66%
Nationality (%)	Chile	96.04%
	Venezuela	0.76%
	Peruvian	0.24%
	Argentina	0.14%
	Bolivia	0.14%
	Colombia	0.14%
	Mexican	0.05%
	Others/variants	<1% each

**Table 2 healthcare-13-01825-t002:** Rotated factor loadings (Varimax) of the exploratory factor analysis.

Article	Factor 1	Factor 2	h2	u2	Complexity
p1	0.66	0.38	0.57	0.43	1.6
p2	0.86	0.34	0.87	0.13	1.3
p3	0.72	0.45	0.71	0.29	1.7
p4	0.48	0.64	0.65	0.35	1.9
p5	0.44	0.73	0.72	0.28	1.6
p6	0.37	0.73	0.67	0.33	1.5
p7	0.34	0.69	0.60	0.40	1.5
p8	0.27	0.59	0.41	0.59	1.4

**Table 3 healthcare-13-01825-t003:** Mean telehealth literacy scores by nationality.

Nationality	N	Mean	SD
Chile	2122	23.5	10.1
Argentina	3	37.7	4.04
Bolivia	4	30.5	6.61
Colombia	4	31.0	6.12
Peruvian	5	30.0	7.84
Venezuela	16	30.1	5.77
Mexican	1	8.0	-
Not specified	45	20.8	11.5

Kruskal–Wallis test: Chi-square = 22.28, df = 13, *p*-value = 0.0022. Telehealth literacy scores range from 8 (minimum) to 40 (maximum) points (sum of eight items rated 1–5).

**Table 4 healthcare-13-01825-t004:** Multiple linear regression predicting continuous telehealth literacy scores.

Predictor	Estimate (β)	Std. Error	t Value	*p*-Value
(Intercept)	37.406	5.417	6.906	<0.001 ***
Age	−0.143	0.012	−11.713	<0.001 ***
Female (ref: Male)	0.762	0.419	1.820	0.069
Frequency of telehealth	0.607	0.051	11.875	<0.001 ***
Rural (ref: Urban)	−0.918	0.509	−1.804	0.071
Bolivian	−5.939	7.123	−0.834	0.405
Colombian	−5.375	7.618	−0.706	0.481
Mexican	−28.781	10.778	−2.670	0.008 **
Peruvian	−2.685	6.820	−0.394	0.694
Venezuelan	−3.213	5.873	−0.547	0.584

Note: *p* < 0.01 (**), *p* < 0.001 (***).

**Table 5 healthcare-13-01825-t005:** Linear regression predicting continuous telehealth literacy score from age, sex, frequency of telehealth use, and urban/rural residence.

	Estimate	Std. Error	t Value	Pr(>|t|)
(Intercept)	28.826	0.651	44.222	<2 × 10^−16^ ***
Age	−0.1452	0.012	−11.870	<2 × 10^−16^ ***
Female (ref: Male)	0.8470	0.418	2.024	0.043 *
Frequency of Telehealth	0.603	0.051	11.803	<2 × 10^−16^ ***
Rural (ref: Urban)	−0.981	0.509	−1.927	0.054

Note: *p* < 0.05 (*), *p* < 0.001 (***).

**Table 6 healthcare-13-01825-t006:** Multinomial logistic regression predicting telehealth literacy levels.

Outcome	Predictor	Estimate (β)	Std. Error	Odds Ratio	95% CI Lower	95% CI Upper	*p*-Value
High	Intercept	23.320	0.355	-	-	-	**<0.001**
High	Age	−0.025	0.010	0.976	0.958	0.994	**0.010**
High	Sex (ref: Male)	0.025	0.328	1.025	0.538	1.950	0.940
High	Frequency of Telehealth	0.655	0.195	1.924	1.312	2.821	**0.001**
High	Rural (ref: Urban)	−2.797	0.388	0.061	0.029	0.131	**<0.001**
High	Bolivian	−4.451	0.463	0.012	0.005	0.029	**<0.001**
High	Colombian	−4.005	0.530	0.018	0.006	0.052	**<0.001**
High	Mexican	−16.131	-	-	-	-	**<0.001**
High	Peruvian	0.975	0.499	2.651	0.996	7.055	0.051
High	Venezuelan	−0.891	0.284	0.410	0.235	0.716	**0.002**
Low	Intercept	0.209	0.311	1.232	0.670	2.268	0.502
Low	Age	0.008	0.010	1.008	0.989	1.027	0.418
Low	Sex (ref: Male)	−0.265	0.333	0.767	0.400	1.472	0.425
Low	Frequency of Telehealth	−0.065	0.204	0.937	0.628	1.397	0.747
Low	Rural (ref: Urban)	−2.926	0.395	0.054	0.028	0.118	**<0.001**
Low	Bolivian	−5.664	~0	0.003	0.003	0.003	**<0.001**
Low	Colombian	−5.642	~0	0.004	0.004	0.004	**<0.001**
Low	Mexican	21.918	~0	-	-	-	**<0.001**
Low	Peruvian	−4.138	~0	0.016	0.016	0.016	**<0.001**
Low	Venezuelan	−4.789	-	0.008	-	-	-
Medium	Intercept	2.967	0.354	19.441	9.715	38.902	**<0.001**
Medium	Age	0.004	0.009	1.004	0.985	1.022	0.705
Medium	Sex (ref: Male)	−0.074	0.326	0.929	0.490	1.759	0.821
Medium	Frequency of Telehealth	0.516	0.195	1.676	1.143	2.458	**0.008**
Medium	Rural (ref: Urban)	−2.479	0.386	0.084	0.039	0.179	**<0.001**
Medium	Bolivian	15.719	0.463	-	-	-	**<0.001**
Medium	Colombian	15.045	0.530	-	-	-	**<0.001**
Medium	Mexican	−5.363	-	0.005	-	-	**<0.001**
Medium	Peruvian	18.927	0.499	-	-	-	**<0.001**
Medium	Venezuelan	17.813	0.284	-	-	-	**<0.001**

Estimated coefficients (β), standard errors (SEs), odds ratios (ORs), 95% confidence intervals (CIs), and *p*-values for each predictor (age, sex, times using telehealth, urban vs. rural, and nationality). “-” indicates that the odds ratio and confidence interval are not reported due to unstable or unreliable estimates, resulting from small samples in certain categories or excessively large standard errors.

**Table 7 healthcare-13-01825-t007:** Reduced multinomial logistic regression predicting telehealth literacy levels by demographics and usage.

Outcome	Predictor	Estimate (β)	Std. Error	Odds Ratio	95% CI Lower	95% CI Upper	*p*-Value
High	(Intercept)	4.836	0.613	—	—	—	<0.001
High	Age	−0.024	0.009	0.976	0.958	0.994	0.010
High	Female (ref: Male)	0.025	0.328	1.026	0.538	1.952	0.938
High	Frequency of Telehealth	0.650	0.194	1.916	1.307	2.809	0.001
High	Rural (ref: Urban)	−2.813	0.388	0.060	0.030	0.127	<0.001
Medium	(Intercept)	3.639	0.611	—	—	—	<0.001
Medium	Age	0.003	0.009	1.003	0.985	1.022	0.730
Medium	Female (ref: Male)	−0.077	0.325	0.926	0.490	1.751	0.811
Medium	Frequency of Telehealth	0.512	0.194	1.670	1.140	2.444	0.009
Medium	Rural (ref: Urban)	−2.490	0.385	0.083	0.039	0.178	<0.001

**Table 8 healthcare-13-01825-t008:** Confusion matrix for the reduced multinomial logistic regression model.

Actual	Low	Medium	High
Low	71	162	191
Medium	74	271	372
High	47	172	690

## Data Availability

Data are unavailable due to privacy and ethical restrictions.

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
