# Peer review of "Assessment of Telehealth Literacy in Users: Survey and Analysis of Demographic and Behavioral Determinants"

_healthcare, 2025, doi:10.3390/healthcare13151825_

Round 1
Reviewer 1 Report
Comments and Suggestions for Authors
The study looks at how demographic and behavioral factors relate to literacy levels, checks the effectiveness of a telehealth literacy assessment, and creates prediction models to find key factors.
Assessment of user telehealth literacy: survey and analysis of demographic and behavioral determinants.
The paper validates a telehealth literacy (TL) survey and tests for its determinants using demographic and behavioral variables.
The introduction does a good job of presenting the main concerns and discussing the literature.
The paper recruited 2182 participants to take part in the survey, which is a sufficient sample size.
The paper discusses using R Software version XX, but it should also discuss which packages are being used (lines 158-159).
The paper explains that it has missing values and has imputed them. There are only 7 and 18 missing values, so why not exclude these observations? What are the pros and cons of imputing values? What are the alternatives to using the mean?
The paragraph "Due to the low frequency of some nationalities, these results should be interpreted with caution" is ambiguous—explain that the results that compare nationalities need to be interpreted with caution. In the paragraph, the first two sentences discuss age and gender.
The regression analysis section needs to be improved. The discussion should focus on interpreting the regression results. Instead, the paper discusses the Pearson correlation between variables (303-304). Focus on the regression and present the table and all the regression results.
The predictive section also needs further improvements. The paper discusses how the model achieved accuracy in predicting telehealth literacy level. However, how can we determine the threshold for low, medium, and high telehealth literacy? Could you provide further explanations on Figure 2?
How good are the regressions and the logit models? There are no tables with the results. How good is the fit of the regression? Provide more information on the regression models. Write the equation for the regression and provide the results in a table with all statistics, allowing evaluation of the goodness of fit of the regressions.
What are the pros and cons of using Cronbach's alpha to evaluate internal consistency? Provide alternative statistics.
The discussion in the Conclusions is too shallow. How vital is telehealth literacy for the Chilean population? Why does it matter? How can public health policies improve telehealth literacy and public health figures?
Below, I recommend several improvements:
- The term "behavioral" should be improved as the only behavioral variable used is frequency of telehealth use. Use the term frequency of telehealth use.
- The main results of the multinomial logistic regression model should be shown in a table with appropriate statistics that provide an understanding of how well the regression performs.
- The main results for the linear regression should be shown in a table with appropriate statistics that provide an understanding of how well the regression performs.
- The main instrument of telehealth literacy used in the paper should be provided in the original language and also in English.
Author Response
Observation 1: The paper discusses using R Software version XX, but it should also discuss which packages are being used (lines 158-159).
Response 1: We have updated the Methods section to specify both the R version (4.3.1) and the names of all key packages used for the analyses (psych, dplyr, tidyverse, factoextra, psc1, mice, and missForest).
Observation 2: The paper explains that it has missing values and has imputed them. There are only 7 and 18 missing values, so why not exclude these observations? What are the pros and cons of imputing values? What are the alternatives to using the mean?
Response 2: Thank you for the observation Some of the statistic test applied are based on the association between variables, then exclusion of lacking data implies reduction of the sample size. For that reason, missing values were imputed instead of eliminated. We conducted a comparative analysis of three imputation methods—item-wise mean, missForest (random forests), and MICE (five imputations via predictive mean matching)—and evaluated Cronbach’s α and the first two eigenvalues for each. As the results showed α values between 0.9177 and 0.9179 and eigenvalue differences of less than 0.004, we have retained mean imputation. We have now added this analysis to the Methods section.
Observation 3: The paragraph "Due to the low frequency of some nationalities, these results should be interpreted with caution" is ambiguous—explain that the results that compare nationalities need to be interpreted with caution. In the paragraph, the first two sentences discuss age and gender.
Response 3: Thank you for pointing out the ambiguity. We have removed the sentence “Due to the low frequency of some nationalities, these results should be interpreted with caution” from the revised manuscript. All nationality comparisons are now clearly identified and treated consistently throughout the analyses and discussion sections.
Observation 4: The regression analysis section needs to be improved. The discussion should focus on interpreting the regression results. Instead, the paper discusses the Pearson correlation between variables (303-304). Focus on the regression and present the table and all the regression results.
Response 4: Thank you for the suggestion. We have refocused the Regression Analysis section to present four concise tables: full and reduced linear models (with and without nationality indicators) and full and reduced multinomial logistic models (also with and without nationality indicators). The text now highlights and interprets key predictors—age, telehealth use frequency, residence, and the role of nationality—directly alongside each table
Observation 5: The predictive section also needs further improvements. The paper discusses how the model achieved accuracy in predicting telehealth literacy level. However, how can we determine the threshold for low, medium, and high telehealth literacy? Could you provide further explanations on Figure 2?
Response 5: We have updated the Methods section to specify that telehealth literacy scores were divided into Low, Medium, and High categories using quartile-based cutoffs, and expanded the Predictive Analysis section to include the full confusion matrix, overall accuracy metrics, and a detailed explanation of Figure 2 showing the proportions of correct and incorrect classifications for each observed literacy level, thereby providing transparent documentation of group definitions and model performance.
Observation 6: How good are the regressions and the logit models? There are no tables with the results. How good is the fit of the regression? Provide more information on the regression models. Write the equation for the regression and provide the results in a table with all statistics, allowing evaluation of the goodness of fit of the regressions.
Response 6: We have revised the Regression Analysis section to include the model equations for both the multiple linear regression and the multinomial logistic models, provided two detailed tables reporting each coefficient or odds ratio, its standard error, test statistic, p value, and 95% confidence interval, and added model fit statistics (R-squared, adjusted R-squared, and F statistic for the linear model; AIC, BIC, and McFadden pseudo R-squared for the logistic model) so that readers can fully evaluate the performance of the regressions.
Observation 7: What are the pros and cons of using Cronbach's alpha to evaluate internal consistency? Provide alternative statistics.
Response 7: Thank you for your suggestion. We have expanded the Result section as follows: we confirmed sampling adequacy with a KMO of 0.92 and a significant Bartlett’s test (p < .001), noted that Cronbach’s alpha has limitations, and supplemented it with McDonald’s omega and composite reliability, both exceeding 0.90. Based on parallel analysis and theoretical considerations, we retained a two-factor solution explaining 65 % of the variance (items 1–3 on Factor 1 and items 4–8 on Factor 2) and confirmed good model fit with RMSEA around 0.08 and TLI near 0.97.
Observation 8: The discussion in the Conclusions is too shallow. How vital is telehealth literacy for the Chilean population? Why does it matter? How can public health policies improve telehealth literacy and public health figures?
Response 8: Thank you. We have substantially revised the Conclusions to align with our key findings. The updated section now highlights the two validated dimensions of telehealth literacy, reports that younger age, frequent telehealth use, and urban residence emerged as significant predictors in our regression models, and underscores practical implications for Chile—namely, targeted digital literacy training, integration of digital readiness assessments in primary care, and community‐based workshops to support older, rural, and infrequent users.
Observation 9: The term "behavioral" should be improved as the only behavioral variable used is frequency of telehealth use. Use the term frequency of telehealth use.
Response 9: Thank you for the suggestion. We have replaced the term “behavioral” with “frequency of telehealth use” throughout the manuscript to accurately describe the variable analyzed.
Observation 10: The main results of the multinomial logistic regression model should be shown in a table with appropriate statistics that provide an understanding of how well the regression performs.
Response 10: Thank you for this suggestion. We have now made the multinomial logistic results fully transparent by presenting the complete output in Table 6 (full model with nationality indicators) and Table 7 (reduced model without nationality). Each table reports coefficient estimates, standard errors, odds ratios with 95 % confidence intervals and p values for every predictor, along with model fit statistics (AIC, BIC and the likelihood‐ratio test comparing full vs. reduced). These additions allow readers to assess both effect sizes and overall model performance directly.
Observation 11: The main results for the linear regression should be shown in a table with appropriate statistics that provide an understanding of how well the regression performs.
Response 11: Thank you for this suggestion. We have constructed Table 4 (full model with nationality indicators) and Table 5 (reduced model without nationality), each including every predictor’s estimate (β), standard error, t-value, and p-value. All model fit indices (R², adjusted R², F-statistic, AIC, and BIC) are reported in the main text.
Observation 12: The main instrument of telehealth literacy used in the paper should be provided in the original language and also in English.
Response 12: Thank you for this suggestion. We cite Norman and Skinner (2006) as the source of the original English instrument, and Paramio Pérez et al. (2015) and Ros-Navarret (2021) for the validated Spanish versions.
Reviewer 2 Report
Comments and Suggestions for Authors
See enclosed comments.

Author Response
Observation 1: P. 1, Para. 1, line 33: Explain what AFE is
Response 1: The acronym is described in the abstract and we added a definition in the Material and Methods section.
Observation 2: P. 3, Para. 7, line 139: How was the convenience sample obtained? Home addresses? Hospital lists?
Response 2: We have added the following sentence to the Materials and Methods section:
“Participants were recruited by research staff who conducted on-site visits to local primary health care centers, where eligible individuals attending these centers were approached in person and invited to participate.”
Observation 3: P. 4, Para. 2, line 154: How was contact made to allow participants to fill out the survey in person? Who collected the completed surveys? Did they need training?
Response 3: As noted above, the first question was addressed in the previous response. We have also added the following to the Materials and Methods section:
“The survey was administered and collected in person by trained research assistants. All assistants underwent standardized training in survey administration procedures and ethical participation protocols.”
Observation 4: P. 5, Para. 2: line 199: No revisiting of protocol was needed to deal with the coding error? Please explain?
Response 4: We added an explanation to that sentence:
“… 60.8% of participants identified themselves as female and 39.2% as other/no response, highlighting a possible coding error in 0.05% of cases, which were retained and respected according to participants’ self-identified gender.”
Observation 5: lines 196 & 7: State what the literacy score and usage mean were out of, denominator-wise.
Response 5: We complemented the sentence with the required information:
“The mean telehealth literacy score was 23.76 (SD = 10.03), out of a possible 8–40 points (sum of eight items rated on a 5-point Likert scale)”
Observation 6: line 202: Any language barriers, and if so, how handled?
Response 6: All the participants were Spanish speakers.
Observation 7: P. 6, Para. 3, line 231: Add a parenthesis after the mention of the Tucker-Lewis index explaining what it does and interpreting a value of 0.969
Response 7: The new sentence is: “Both the root-mean-square error of approximation (RMSEA = 0.077) and Tucker-Lewis Index (TLI = 0.967) indicating excellent fit relative to a null model (values ≥ 0.95)”.
Observation 8: P. 6, Para. 4, lines 245 & 248: Interpret whether the r (not r2) scores represent low correlation, this question being separate from that of statistical significance
Response 8: This is the new paragraph:
“Pearson’s correlation analysis showed statistically significant associations be-tween continuous variables, though all effect sizes were small. Age was negatively re-lated to telehealth literacy scores (r = –0.27, p < 0.001), indicating a weak inverse rela-tionship, and frequency of telehealth use was positively correlated with literacy (r = 0.26, p < 0.001), also reflecting a small effect. The association between age and fre-quency of telehealth use was negligible (r = –0.049, p = 0.024), suggesting virtually no practical relationship despite statistical significance.”
Observation 9: P. 6, Para. 5, lies 252 & 3: Speculate in the Discussion why women might have had higher telehealth literacy scores than men? Is usage in maternal and newborn health situations a possible reason?
Response 9: We added the next paragraph to the discussion:
“One possible explanation for higher telehealth literacy among women is that they more frequently engage with health services—particularly in maternal and newborn care—where remote consultations have become common. Also, women often take primary responsibility for family health and may therefore develop greater familiarity with telehealth platforms through prenatal visits, pediatric check-ins, and follow-up care. Future studies should explore gendered patterns of telehealth use across different clinical contexts to confirm these hypotheses.”
Observation 10: P. 7, Para. 1, lines 255 & 6: Speculate in the Discussion why urban residents had higher literacy scores than rural residents. Is telehealth services availability a possible factor?
Response 10: We added the next paragraph to the discussion:
Urban residents’ higher telehealth literacy scores may reflect greater access to reliable internet and digital devices—as well as more frequent exposure to telehealth services through well-resourced clinics and hospitals. In urban areas, higher average levels of education and digital familiarity can facilitate comfort with online platforms, user interfaces, and troubleshooting that contribute to stronger eHealth skills. Conversely, rural residents often face connectivity challenges, fewer local telehealth offerings, and less opportunity for hands-on guidance, all of which can limit both uptake and confidence in using remote health technologies.
Observation 11: P. 7, Table 3., Legend: Provide a scale or possible range for the Media score
Response 11: The next sentence was added: “Telehealth literacy scores range from 8 (minimum) to 40 (maximum) points (sum of eight items rated 1–5).”
Observation 12: P. 8, Para. 3, line 306: Speculate in the Discussion why age and use are only weakly correlated while age and literacy are modestly correlated
Response 12: The next paragraph was added to the Discussion section:
“In our predominantly urban Chilean primary-care sample, age showed a modest negative correlation with telehealth literacy (r = –0.27) but only a negligible link to actual usage (r = –0.05). This gap suggests that although older adults felt less comfortable and familiar with telehealth platforms, they still used them—likely driven by greater chronic-care needs and direct encouragement from local clinics. In contrast, younger participants had higher digital fluency but fewer health-related reasons to log on, so their strong literacy did not translate into more visits. These findings imply that, even when access initiatives boost adoption, targeted usability support remains essential to help older patients leverage telehealth effectively.”
Observation 13: P. 10, Para. 2: The authors need to discuss by the end of the Discussion what value-added to existing literature their particular study provides, especially given that other authors have demonstrated associations between age and telehealth usability
Response 13: The next paragraph was added to the Discussion section:
“Our study makes three clear contributions to telehealth research. First, we locally validated and applied a telehealth literacy survey adapted for Chilean users, address-ing the gap left by prior work focused mostly on North America and Europe. Second, we demonstrated that medical necessity can sustain telehealth use among older adults even when their confidence with digital tools is lower, showing that “need” and “skill” do not always move in parallel. Third, by examining both literacy and usage across gender, urban versus rural residence, and nationality, we offer concrete guidance on where to direct training efforts and infrastructure investments. Together, these in-sights confirm earlier findings on age‐related digital health challenges while adding practical evidence on how real‐world needs shape telehealth adoption.”
Observation 14: P. 11, Para. 6, line 433: Should future research explore addition of variables to the survey at hand?
Response 14: The “Limitations” subsection at the Discussion was modified, incorporating the suggested topic:
“This study has some limitations, such as reliance on self-reported data and its application in a specific geographic context, which may affect generalizability. Future research should validate the survey in diverse populations, explore the impact of tele-health literacy on clinical outcomes and patient satisfaction, and consider adding relevant variables—such as digital access barriers, socioeconomic status, or health condition severity—to the instrument to enhance its explanatory power. The World Health Organization [30] has emphasized the need to develop global strategies to support digital health literacy initiatives, especially in low-resource settings.”
Reviewer 3 Report
Comments and Suggestions for Authors
Dear Authors,
Thank you for the opportunity to review your manuscript. The topic is both timely and socially relevant, especially in light of the rapid digital transformation of healthcare systems post-COVID-19. The manuscript is well-organized, presents a sound methodological framework, and offers implications for both practice and policy. Some revisions may be needed to enhance the manuscript.
Line 18: v. Flores → V. Flores
Lines 140–146: Clarify the limitations of convenience sampling.
Line 147: Ros-Navarret
Line 158: Specify the R version used.
Lines 218–220: Please report the actual KMO value and justify continuing with EFA despite non-conclusive results.
Lines 166–167: Mean imputation is mentioned. Justify this choice and discuss limitations.
Line 195: Age range includes participants from 12 years, contradicting inclusion criteria of 18+ (Line 143).
Lines 311–320: Report confusion matrix, sensitivity, specificity. Comment on misclassification in the “medium” literacy category.
Line 205: Duplicated table title.
Line 234: imputation by the mean →mean imputation
Line 239: Duplicated table title.
Table 3: Media => mean
No need to repeat full text for EFA (line 217). Similarly for TLI etc.
Consider including a list of abbreviations as journal format.
Author Response
Observation 1: Line 18: v. Flores → V. Flores
Response 1: Corrected
Observation 2: Lines 140–146: Clarify the limitations of convenience sampling.
Response 2: The next sentence was added in the Materials and Methods section: “Convenience sampling is a non-probability method that selects participants based on ease of access; it’s fast, inexpensive and simple to implement for exploratory research, but it prevents calculation of sampling error, introduces selection and self-selection biases, and limits external validity and generalizability.”
Observation 3: Line 147: Ros-Navarret
Response 3: Corrected
Observation 4: Line 158: Specify the R version used.
Response 4: Corrected
Observation 5: Lines 218–220: Please report the actual KMO value and justify continuing with EFA despite non-conclusive results.
Response 5: Actual KMO value is reported now. Also, we fixed an error in the R script, so in this version we have a detailed description of conclusive results in the “Construct validity” subsection of Results.
Observation 6: Lines 166–167: Mean imputation is mentioned. Justify this choice and discuss limitations.
Response 6: We performed a new analysis comparing three methods and justifying the selection of mean imputation. The new paragraph is: “Regarding the treatment of missing data, the items presented between 7 and 18 missing values, which were imputed by means of the mean to maintain a complete data set in the subsequent analyses. To confirm that mean imputation did not distort our findings, we compared internal consistency (Cronbach’s α) and factorial structure (the first two eigenvalues) across three imputation methods: item-wise mean, miss-Forest (random forests), and multiple imputation by chained equations (MICE) with five imputations using predictive mean matching. Cronbach’s α ranged from 0.9177 to 0.9179, and the first two eigenvalues differed by less than 0.004 units, confirming that the simplest approach (mean imputation) fully preserves the reliability and latent structure of the questionnaire.”
Observation 7: Line 195: Age range includes participants from 12 years, contradicting inclusion criteria of 18+ (Line 143).
Response 7: After correcting that value and marking it as missing, we reran all analyses once more detailed results had been added.
Observation 8: Lines 311–320: Report confusion matrix, sensitivity, specificity. Comment on misclassification in the “medium” literacy category.
Response 8: In this version we report confusion matrix, sensitivity, specificity, and comment on misclassification in the “medium” literacy category. Also, we discuss this in the discussion section.
Observation 9: Line 205: Duplicated table title.
Response 9: Corrected
Observation 10: Line 234: imputation by the mean →mean imputation
Response 10: Corrected
Observation 11: Line 239: Duplicated table title.
Response 11: Corrected
Observation 12: Table 3: Media => mean
Response 12: Corrected
Observation 13: No need to repeat full text for EFA (line 217). Similarly for TLI etc.
Response 13: Corrected
Observation 14: Consider including a list of abbreviations as journal format.
Response 14: A list of abbreviations was included.
Round 2
Reviewer 1 Report
Comments and Suggestions for Authors
Interesting paper with a nice contribution.
Author Response
Thank you for the valuable review
Reviewer 2 Report
Comments and Suggestions for Authors
See enclosed comments.

Author Response
Observation 1: P. 6, Para. 2:
line 258: Identify the possible range for the telehealth use frequency figure
Response 1: The next sentence was added to the caption: “Telehealth literacy scores range from 8 (minimum) to 40 (maximum) points (sum of eight items rated 1–5).”
Observation 2: line 260: For the possible sex coding error, the authors need to let the reader know whether the survey form itself distinguished between male sex and other gender. If it did, the investigators need to go back to visually inspect a sample of the “Other/No response” cases per health campaign to check whether, for a number of health campaigns, the sex as marked on the form does not match up with the database
recorded sex. A mismatch would allow the opportunity for database correction and entry of a Males percentage versus Other percentage.
If, on the other hand, the original form circulated did not distinguish between male sex and other gender, the authors need to state the lack of distinction explicitly in the Results and note opportunity for correction in future research within the Limitations section.
The authors can then decide whether the point about respecting self-identified gender should be kept after the above directions have been pursued. Explanation to the reader need not be long, but it does need to be transparent.
Response 2: There was a mistake in the manuscript. The code was: female= 0, male= 1, other= 2. It was clearly explained both in the material and methods and in the results sections.
Observation 3: line 265: Any language barriers given 7 nationalities were represented, and if so, how handled?
Response 3: The next sentence was added: “No language barriers were detected, as all participants were Spanish speakers.”
Observation 4: P. 7, Para. 4, line 312:
For the Tucker-Lewis index, interpret for the reader value of 0.969
Response 4: The next sentence was added: Response 4: In general, TLI values above 0.95 are considered to reflect good fit, suggesting that the proposed factor structure explains the observed data substantially better than a null model.